# TransMIL: Transformer based Correlated Multiple Instance Learning for Whole Slide Image Classification

**Zhuchen Shao**[*,1], **Hao Bian**[*,1], **Yang Chen**[*,1], **Yifeng Wang**[2], **Jian Zhang**[3], **Xiangyang Ji**[4]
**Yongbing Zhang**[†,2]

[1]Tsinghua Shenzhen International Graduate School, Tsinghua University
[2]Harbin Institute of Technology (Shenzhen)
[3]School of Electronic and Computer Engineering, Peking University
[4]Department of Automation, Tsinghua University

## Abstract

Multiple instance learning (MIL) is a powerful tool to solve the weakly supervised classification in whole slide image (WSI) based pathology diagnosis. However, the current MIL methods are usually based on independent and identical distribution hypothesis, thus neglect the correlation among different instances. To address this problem, we proposed a new framework, called correlated MIL, and provided a proof for convergence. Based on this framework, we devised a Transformer based MIL (TransMIL), which explored both morphological and spatial information. The proposed TransMIL can effectively deal with unbalanced/balanced and binary/multiple classification with great visualization and interpretability. We conducted various experiments for three different computational pathology problems and achieved better performance and faster convergence compared with state-of-the-art methods. The test AUC for the binary tumor classification can be up to 93.09% over CAMELYON16 dataset. And the AUC over the cancer subtypes classification can be up to 96.03% and 98.82% over TCGA-NSCLC dataset and TCGA-RCC dataset, respectively. Implementation is available at: https://github.com/szc19990412/TransMIL.

## 1 Introduction

The advent of whole slide image (WSI) scanners, which convert the tissue on the biopsy slide into a gigapixel image fully preserving the original tissue structure [1], provides a good opportunity for the application of deep learning in the field of digital pathology [2, 3, 4]. However, the deep learning based biopsy diagnosis in WSI has to face a great challenges due to the huge size and the lack of pixel-level annotations[5]. To address this problem, multiple instance learning (MIL) is usually adopted to take diagnosis analysis as a weakly supervised learning problem.

In deep learning based MIL, one straightforward idea is to perform pooling operation [6, 7] on instance feature embeddings extracted by CNN. Ilse et al. [8] proposed an attention based aggregation operator, giving each instance additional contribution information through trainable attention weights. In addition, Li et al. [9] introduced non-local attention into the MIL problem. By calculating the similarity between the highest-score instance and the others, each instance is given different

---

[*]Contributed equally: {shaozc0412,h2495067728,sky374263410}@gmail.com.
[†]Corresponding author: ybzhang08@hit.edu.cn.

35th Conference on Neural Information Processing Systems (NeurIPS 2021).

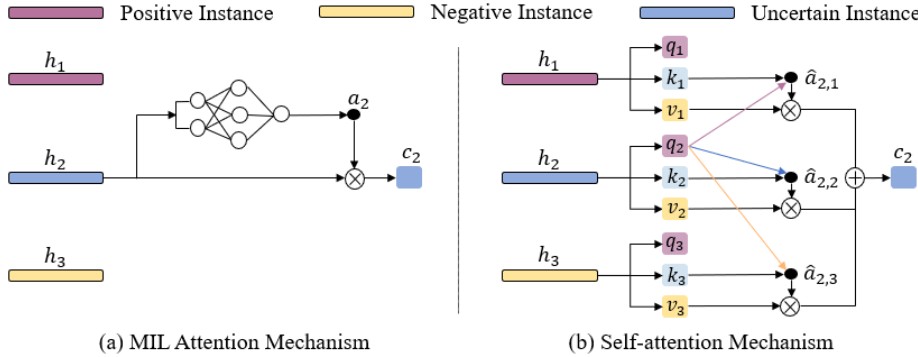

Figure 1: Decision-making process. MIL Attention Mechanism: follow the i.i.d. assumption. Self-attention Mechanism: under the correlated MIL framework.

attention weight and the interpretable attention map can be obtained accordingly. There were also other pioneering works [10, 11, 12, 13, 14] in weakly supervised WSI diagnosis.

However, all these methods are based on the assumption that all the instances in each bag are independent and identically distributed (i.i.d.). While achieving some improvements in many tasks, this i.i.d. assumption was not entirely valid [15] in many cases. Actually, pathologists often consider both the contextual information around a single area and the correlation information between different areas when making a diagnostic decision. Therefore, it would be much desirable to consider the correlation between different instances in MIL diagnosis.

At present, Transformer is widely used in many vision tasks [16, 17, 18, 19, 20, 21] due to the strong ability of describing correlation between different segments in a sequence (tokens) as well as modelling long distance information. As shown in Figure 1, different from bypass attention network in the existing MIL, the Transformer adopts self-attention mechanism, which can pay attention to the pairwise correlation between each token within a sequence. However, traditional Transformer sequences are limited by their computational complexity and can only tackle shorter sequences (e.g., less than 1000) [22]. Therefore, it is not suitable for large size images such as WSIs.

To address these challenges mentioned above, we proposed a correlated MIL framework, including the convergence proof and a generic three-step algorithm. In addition, a Transformer based MIL (TransMIL) was devised to explore both morphological and spatial information between different instances. Great performance over various datasets demonstrate the validity of the proposed method.

## 2 Related Work

### 2.1 Application of MIL in WSI classification

The application of MIL in WSIs can be divided into two categories. The first one is instance-level algorithms [7, 23, 24, 25, 26], where a CNN is first trained by assigning each instance a pseudo-label based on the bag-level label, and then the top-k instances are selected for aggregation. However, this method requires a large number of WSIs, since only a small number of instances within each slide can actually participate in the training. The second category is embedding-level algorithms, where each patch in the entire slide is mapped to a fixed-length embedding, and then all feature embeddings are aggregated by an operator (e.g., max-pooling). To improve the performance, the MIL attention based method [8, 10, 11, 12, 13] assigns the contribution of each instance by introducing trainable parameters. In addition, the feature clustering methods [14, 27, 28] calculated the cluster centroids of all the feature embeddings and then the representative feature embeddings was employed to make the final prediction. Recently, non-local attention [9] was also adopted in MIL to pay more attention to the correlation between the highest-score instance and all the remaining instances.

## 2.2 Attention and Self-attention in Deep Learning

Attention was initially used to extract important information about sentences in machine translation [29]. Then the attention mechanism was gradually applied to computer vision tasks, including giving different weights to feature channels [30] or spatial distribution [31], or giving different weights to time series in video analysis [32]. Recently, attention was also applied in MIL analysis [10, 11, 12, 13]. However, all these methods did not consider the correlation between different instances.

The most typical self-attention application was the Transformer based NLP framework proposed by Google [33]. Recently, Transformer was also applied in many computer vision tasks, including object detection [16, 17], segmentation [18, 19], image enhancement [20, 21] and video processing [34]. In this paper, for the first time, we proposed a Transformer based WSI classification, where the correlations among different instances within the same bag are comprehensively considered.

## 3 Method

### 3.1 Correlated Multiple Instance Learning

**Problem formulation**   Take binary MIL classification as an example, we want to predict a target value $Y_i \in \{0, 1\}$, given a bag of instances $\{\boldsymbol{x}_{i,1}, \boldsymbol{x}_{i,2}, \ldots, \boldsymbol{x}_{i,n}\}$ with $\mathbf{X}_i$, for $i = 1, \ldots, b$, that exhibit both dependency and ordering among each other. The instance-level labels $\{y_{i,1}, y_{i,2}, \ldots, y_{i,n}\}$ are unknown, and the bag-level label is $Y_i$, for $i = 1, \ldots, b$. A binary MIL classification can be defined as:

$$Y_i = \begin{cases} 0, \text{ iff } \sum y_{i,j} = 0 & y_{i,j} \in \{0, 1\}, j = 1 \ldots n \\ 1, \text{ otherwise} \end{cases} \tag{1}$$

$$\hat{Y}_i = S(\mathbf{X}_i), \tag{2}$$

where $S$ is a scoring function, $\hat{Y}_i$ represents the prediction. $b$ is the total number of bags, $n$ is the number of instances in $i$th bag, and the number of $n$ can vary for different bags.

Compared to the MIL framework proposed by Ilse *et al.* [8], we further introduce the correlation between different instances. Theorem 1 and Inference give an arbitrary approximation form of the scoring function $S(\mathbf{X})$, and Theorem 2 provides the advantage of correlated MIL.

**Theorem 1.**   *Suppose $S : \mathcal{X} \rightarrow \mathbb{R}$ is a continuous set function w.r.t Hausdorff distance $d_H(\cdot, \cdot)$. $\forall \varepsilon > 0$, for any invertible map $P : \mathcal{X} \rightarrow \mathbb{R}^n, \exists$ function $\sigma$ and $g$, such that for any $\mathbf{X} \in \mathcal{X}$:*

$$|S(\mathbf{X}) - g(\underset{\mathbf{X} \in \mathcal{X}}{P} \{\sigma(\boldsymbol{x}) : \boldsymbol{x} \in \mathbf{X}\})| < \varepsilon. \tag{3}$$

*That is: a Hausdorff continuous function $S(\mathbf{X})$ can be arbitrarily approximated by a function in the form $g(\underset{\mathbf{X} \in \mathcal{X}}{P} \{\sigma(\boldsymbol{x}) : \boldsymbol{x} \in \mathbf{X}\})$.*

*Proof.*   By the continuity of $S$, we take $\forall \varepsilon > 0, \exists \delta_\varepsilon$, so that $|S(\mathbf{X}) - S(\mathbf{X}')| < \varepsilon$ for any $\mathbf{X}, \mathbf{X}' \in \mathcal{X}$, if $d_H(\mathbf{X}, \mathbf{X}') < \delta_\varepsilon$.

Define $K = \lceil \frac{1}{\delta_\varepsilon} \rceil$ and define an auxiliary function: $\sigma(\boldsymbol{x}) = \frac{\lfloor K\boldsymbol{x} \rfloor}{K}$. Let $\tilde{\mathbf{X}} = \{\sigma(\boldsymbol{x}) : \boldsymbol{x} \in \mathbf{X}\}$, then:

$$|S(\mathbf{X}) - S(\tilde{\mathbf{X}})| < \varepsilon, \tag{4}$$

because $d_H(\mathbf{X}, \tilde{\mathbf{X}}) < \frac{1}{K} \leq \delta_\varepsilon$.

Let $P : \mathcal{X} \rightarrow \mathbb{R}^n$ be any invertible map, its inverse mapping is expressed as $P^{-1}: \mathbb{R}^n \rightarrow \mathcal{X}$. Let $g = S(P^{-1})$, then:

$$S\left(P^{-1}(\underset{\mathbf{X} \in \mathcal{X}}{P}(\{\sigma(\boldsymbol{x}) : \boldsymbol{x} \in \mathbf{X}\}))\right) = S\left(P^{-1}(\underset{\tilde{\mathbf{X}} \in \mathcal{X}}{P}(\tilde{\mathbf{X}}))\right) = S(\tilde{\mathbf{X}}). \tag{5}$$

Because $|S(\mathbf{X}) - S(\tilde{\mathbf{X}})| < \varepsilon$ and $S(\tilde{\mathbf{X}}) = S\left(P^{-1}(P(\tilde{\mathbf{X}}))\right) = g(P(\tilde{\mathbf{X}}))$, we have:

$$|S(\mathbf{X}) - g(\underset{\mathbf{X} \in \mathcal{X}}{P} \{\sigma(\boldsymbol{x}) : \boldsymbol{x} \in \mathbf{X}\})| < \varepsilon. \tag{6}$$

This completes the proof.  □

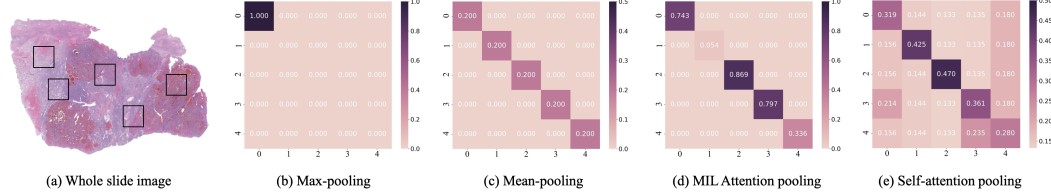

(a) Whole slide image    (b) Max-pooling    (c) Mean-pooling    (d) MIL Attention pooling    (e) Self-attention pooling

Figure 2: The difference between different Pooling Matrix $\mathbf{P}$. Suppose there are 5 instances sampled from WSI in (a), $\mathbf{P} \in \mathbb{R}^{5 \times 5}$ is the corresponding Pooling Matrix, where the values in the diagonal line indicate the attention weight for itself and the rest indicate correlation between different instances. (b,c,d) all neglect the correlation information, hence the $\mathbf{P}$ is diagonal matrix. In (b), the first instance was chosen by Max-pooling operator, so there is only one non-zero value in the first diagonal position. In (c), all the values within diagonal line are the same due to the Mean-pooling operator. In (d), the values within diagonal line can be varied due to the introduction of bypass attention. (e) obeys the correlation assumption, so there are non-zero values in off-diagonal position indicating correlation between different instances.

**Inference** *Suppose $S : \mathcal{X} \to \mathbb{R}$ is a continuous set function w.r.t Hausdorff distance $d_H(\cdot, \cdot)$. $\forall \varepsilon > 0$, for any function $f$ and any invertible map $P : \mathcal{X} \to \mathbb{R}^n, \exists$ function $h$ and $g$, such that for any $\mathbf{X} \in \mathcal{X}$:*

$$|S(\mathbf{X}) - g(\underset{\mathbf{X} \in \mathcal{X}}{P}\{f(\boldsymbol{x}) + h(\boldsymbol{x}) : \boldsymbol{x} \in \mathbf{X}\})| < \varepsilon. \tag{7}$$

*That is: a Hausdorff continuous function $S(\mathbf{X})$ can be arbitrarily approximated by a function in the form $g(\underset{\mathbf{X} \in \mathcal{X}}{P}\{f(\boldsymbol{x}) + h(\boldsymbol{x}) : \boldsymbol{x} \in \mathbf{X}\})$.*

*Proof.* The proof is in the Appendix A. □

**Theorem 2.** *The Instances in the bag are represented by random variables $\Theta_1, \Theta_2, \ldots, \Theta_n$, the information entropy of the bag under the correlation assumption can be expressed as $H(\Theta_1, \Theta_2, \ldots, \Theta_n)$, and the information entropy of the bag under the i.i.d. (independent and identical distribution) assumption can be expressed as $\sum_{t=1}^{n} H(\Theta_t)$, then we have:*

$$H(\Theta_1, \Theta_2, \ldots, \Theta_n) = \sum_{t=2}^{n} H(\Theta_t \mid \Theta_1, \ldots, \Theta_{t-1}) + H(\Theta_1) \leq \sum_{t=1}^{n} H(\Theta_t). \tag{8}$$

*Proof.* The proof is in the Appendix A. □

Theorem 2 proved the correlation assumption has smaller information entropy, which may reduce the uncertainty and bring more useful information for the MIL problem. Motivated by the Inference and Theorem 2, a generic three-step method like Algorithm1 was developed. The main difference between the proposed algorithm and existing methods is shown in Figure 2.

---

**Algorithm 1:** A generic three-step approach under the correlated MIL

---

**Input:** The bag of instances $\mathbf{X}_i = \{\boldsymbol{x}_{i,1}, \boldsymbol{x}_{i,2} \ldots, \boldsymbol{x}_{i,n}\}$

**Output:** Bag-level predicted label $\hat{Y}_i$

    1) Extracting morphological and spatial information of all the instances by $f$ and $h$, respectively;

    $\mathbf{X}_f \leftarrow f(\mathbf{X}_i), \mathbf{X}_h \leftarrow h(\mathbf{X}_i), \mathbf{X}_{fh} \leftarrow \mathbf{X}_f + \mathbf{X}_h$, where $\mathbf{X}_f, \mathbf{X}_h, \mathbf{X}_{fh} \in \mathbb{R}^{n \times d}$;

    2) Aggregating the extracted information for all instances by Pooling Matrix $\mathbf{P}$;

    $\mathbf{X}_P \leftarrow \mathbf{P}\mathbf{X}_{fh}$, where $\mathbf{P} \in \mathbb{R}^{n \times n}$;

    3) Transforming $\mathbf{X}_P$ to obtain the predicted bag-level label by $g$;

    $\hat{Y}_i \leftarrow g(\mathbf{X}_P)$.

---

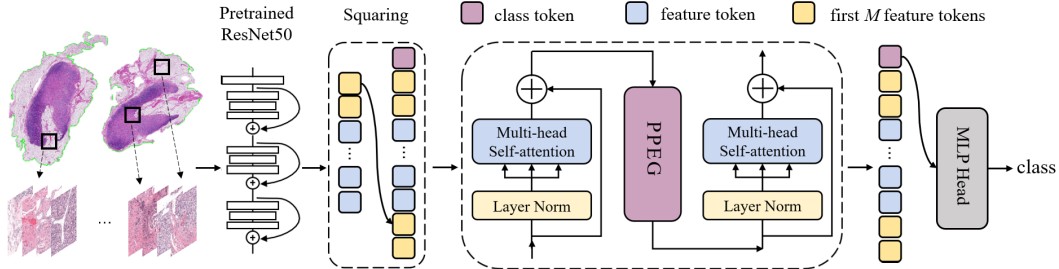

Figure 3: Overview of our TransMIL. Each WSI is cropped into patches (background is discarded), and embedded in feature vectors by ResNet50. Then the sequence is processed with the TPT module: 1) Squaring of sequence; 2) Correlation modelling of the sequence; 3) Conditional position encoding and local information fusion; 4) Deep feature aggregation; 5) Mapping of $\mathbb{T} \rightarrow \mathcal{Y}$.

## 3.2 How to apply Transformer to correlated MIL

The Transformer uses a self-attention mechanism to model the interactions between all tokens in a sequence, and the adding of positional information further increases the use of sequential order information. Therefore it's a good idea to introduce the Transformer into the correlated MIL problem where the function $h$ encodes the spatial information among instances, and the Pooling Matrix $\mathbf{P}$ uses self-attention for information aggregation. To make this clear, we further give a formal denition.

**Transformer based MIL.** Given a set of bags $\{\mathbf{X}_1, \mathbf{X}_2, \ldots, \mathbf{X}_b\}$, and each bag $\mathbf{X}_i$ contains multiple instances $\{\boldsymbol{x}_{i,1}, \boldsymbol{x}_{i,2}, \ldots, \boldsymbol{x}_{i,n}\}$ and a corresponding label $Y_i$. The goal is to learn the mappings: $\mathbb{X} \rightarrow \mathbb{T} \rightarrow \mathcal{Y}$, where $\mathbb{X}$ is the bag space, $\mathbb{T}$ is the Transformer space and $\mathcal{Y}$ is the label space. The specific mapping form of $\mathbb{X} \rightarrow \mathbb{T}$ and $\mathbb{T} \rightarrow \mathbb{Y}$ are available in the Appendix B.

## 3.3 TransMIL for Weakly Supervised WSI Classication

To better describe the mapping of $\mathbb{X} \rightarrow \mathbb{T}$, we design a TPT module with two Transformer layers and a position encoding layer, where Transformer layers are designed for aggregating morphological information and Pyramid Position Encoding Generator (PPEG) is designed for encoding spatial information. The overview of proposed Transformer based MIL (TransMIL) is shown in Figure 3.

**Long Instances Sequence Modelling with TPT.** The sequences are from the feature embeddings in each WSI. The processing steps of the TPT module are shown in Algorithm 2, where MSA denotes Multi-head Self-attention, MLP denotes Multilayer Perceptron, and LN denotes Layer Norm.

---

**Algorithm 2:** TPT module processing flow

---

**Input:** A bag of feature embeddings $\mathbf{H}_i = \{\boldsymbol{h}_{i,1}, \ldots, \boldsymbol{h}_{i,n}\}$, where $\boldsymbol{h}_{i,j} \in \mathbb{R}^{1 \times d}$ is the embedding of the $j$th instance, $\mathbf{H}_i \in \mathbb{R}^{n \times d}$

**Output:** Bag-level predicted label $\hat{Y}_i$

    1) Squaring of sequence;

    $\sqrt{N} \leftarrow \lceil \sqrt{n} \rceil$, $M \leftarrow N - n$, $\mathbf{H}_S \leftarrow \text{Concat}(\boldsymbol{h}_{i,class}, \mathbf{H}_i, (\boldsymbol{h}_{i,1}, \ldots, \boldsymbol{h}_{i,M}))$, where $\boldsymbol{h}_{i,class} \in \mathbb{R}^{1 \times d}$ represents class token, $\mathbf{H}_S \in \mathbb{R}^{(N+1) \times d}$;

    2) Correlation modelling of the sequence;

    $\mathbf{H}_S^\ell \leftarrow \text{MSA}(\mathbf{H}_S)$, where $\ell$ denotes the layer index of the Transformer, $\mathbf{H}_S^\ell \in \mathbb{R}^{(N+1) \times d}$;

    3) Conditional position encoding and local information fusion;

    $\mathbf{H}_S^P \leftarrow \text{PPEG}(\mathbf{H}_S^\ell)$, where $\mathbf{H}_S^P \in \mathbb{R}^{(N+1) \times d}$;

    4) Deep feature aggregation;

    $\mathbf{H}_S^{\ell+1} \leftarrow \text{MSA}(\mathbf{H}_S^P)$, where $\mathbf{H}_S^{\ell+1} \in \mathbb{R}^{(N+1) \times d}$;

    5) Mapping of $\mathbb{T} \rightarrow \mathcal{Y}$;

    $\hat{Y}_i \leftarrow \text{MLP}\left(\text{LN}\left((\mathbf{H}_S^{\ell+1})^{(0)}\right)\right)$, where $(\mathbf{H}_S^{\ell+1})^{(0)} \in \mathbb{R}^{1 \times d}$ represents class token.

---

For most cases, the softmax used in Transformer for vision tasks such as [17, 18, 35] is a row-by-row softmax normalization function. The standard self-attention mechanism requires the calculation of similarity scores between each pair of tokens, resulting in both memory and time complexity of $O(n^2)$. To deal with the long instances sequence problem in WSIs, the softmax in TPT adopts the Nystrom Method proposed in [22]. The approximated self-attention form $\hat{\mathbf{S}}$ can be defined as:

$$\hat{\mathbf{S}} = \mathrm{softmax}\left(\frac{\mathbf{Q}\tilde{\mathbf{K}}^T}{\sqrt{d_q}}\right)\left(\mathrm{softmax}\left(\frac{\tilde{\mathbf{Q}}\tilde{\mathbf{K}}^T}{\sqrt{d_q}}\right)\right)^+ \mathrm{softmax}\left(\frac{\tilde{\mathbf{Q}}\mathbf{K}^T}{\sqrt{d_q}}\right), \qquad (9)$$

where $\tilde{\mathbf{Q}}$ and $\tilde{\mathbf{K}}$ are the $m$ selected landmarks from the original $n$ dimensional sequence of $\mathbf{Q}$ and $\mathbf{K}$, and $\mathbf{A}^+$ is a Moore-Penrose pseudoinverse of $\mathbf{A}$. The final computational complexity is reduced from $O(n^2)$ to $O(n)$. By doing this, the TPT module with approximation processing can satisfy the case where a bag contains thousands of tokens as input.

**Position encoding with PPEG.** In WSIs, the number of tokens in the corresponding sequence often varies due to the inherently variable size of the slide and tissue area. In [36] it is shown that the adding of zero padding can provide an absolute position information to convolution. Inspired by this, we designed the PPEG module accordingly. The overview is shown in Figure 4, and the pseudo-code for the processing is shown in Algorithm 3.

Figure 4: Pyramid Position Encoding Generator. 1) The sequence is divided into patch tokens and class token; 2) Patch tokens are reshaped into 2-D image space; 3) Different sized convolution kernels are used to encode spatial information; 4) Different spatial information are fused together; 5) Patch tokens are flattened into sequence; 6) Connect class token and patch tokens.

Our PPEG module has more advantages over the method proposed in [37]: (1) PPEG module uses different sized convolution kernels in the same layer, which can encode the positional information with different granularity, enabling high adaptability of PPEG. (2) Taking advantage of CNN's ability to aggregate context information, the tokens in the sequence is able to obtain both global information and context information, which enriches the features carried by each token.

---

**Algorithm 3:** PPEG processing flow

---

**Input:** A bag of feature embeddings $\mathbf{H}_S^{\ell}$ after correlation modelling, where $\mathbf{H}_S^{\ell} \in \mathbb{R}^{(N+1)\times d}$.
**Output:** The feature embeddings $\mathbf{H}_S^P$ after conditional position encoding and local information
 fusion, where $\mathbf{H}_S^P \in \mathbb{R}^{(N+1)\times d}$.
 1) Split: $\mathbf{H}_S^{\ell}$ is divided into patch tokens $\mathbf{H}_f$ and class token $\mathbf{H}_c$;
 $\mathbf{H}_f, \mathbf{H}_c \leftarrow \mathrm{Split}\left(\mathbf{H}_S^{\ell}\right)$, where $\mathbf{H}_f \in \mathbb{R}^{N\times d}, \mathbf{H}_c \in \mathbb{R}^{1\times d}$;
 2) Spatial Restore: patch tokens $\mathbf{H}_f$ are reshaped to $\mathbf{H}_S^f$ in the 2-D image space;
 $\mathbf{H}_S^f \leftarrow \mathrm{Restore}\left(\mathbf{H}_f\right)$, where $\mathbf{H}_S^f \in \mathbb{R}^{\sqrt{N}\times\sqrt{N}\times d}$;
 3) Group Convolution: using a set of group convolutions with kernel $k$ and $\frac{k-1}{2}$ zero
paddings$(k=3,5,7)$ to obtain $\mathbf{H}_t^f, t=1,2,3$;
 $\mathbf{H}_t^f \leftarrow \mathrm{Conv}\left(\mathbf{H}_S^f\right)$, where $\mathbf{H}_t^f \in \mathbb{R}^{\sqrt{N}\times\sqrt{N}\times d}, t=1,2,3$;
 4) Fusion: $\mathbf{H}_S^f$ and the $\mathbf{H}_t^f, t=1,2,3$ obtained from the convolution block processing are
added together to obtain $\mathbf{H}_S^F$;
 $\mathbf{H}_S^F \leftarrow \mathbf{H}_S^f + \mathbf{H}_1^f + \mathbf{H}_2^f + \mathbf{H}_3^f$, where $\mathbf{H}_S^F \in \mathbb{R}^{\sqrt{N}\times\sqrt{N}\times d}$;
 5) Flatten: $\mathbf{H}_S^F$ are flattened into sequence $\mathbf{H}_{se}$;
 $\mathbf{H}_{se} \leftarrow \mathrm{Flatten}\left(\mathbf{H}_S^F\right)$, where $\mathbf{H}_{se} \in \mathbb{R}^{N\times d}$;
 6) Concat: connect $\mathbf{H}_{se}$ and class token $\mathbf{H}_c$ to obtain $\mathbf{H}_S^P$;
 $\mathbf{H}_S^P \leftarrow \mathrm{Concat}\left(\mathbf{H}_{se}, \mathbf{H}_c\right)$, where $\mathbf{H}_S^P \in \mathbb{R}^{(N+1)\times d}$.

---

# 4   Experiments and Results

To demonstrate the superior performance of the proposed TransMIL, various experiments were conducted over three public datasets: CAMELYON16, The Caner Genome Atlas (TCGA) non-small cell lung cancer (NSCLC), as well as the TCGA renal cell carcinoma (RCC).

**Dataset.**   CAMELYON16 is a public dataset for metastasis detection in breast cancer, including 270 training sets and 130 test sets. After pre-processing, a total of about 3.5 million patches at ×20 magnification, in average about 8,800 patches per bag were obtained.

TCGA-NSCLC includes two subtype projects, i.e., Lung Squamous Cell Carcinoma (TGCA-LUSC) and Lung Adenocarcinoma (TCGA-LUAD), for a total of 993 diagnostic WSIs, including 507 LUAD slides from 444 cases and 486 LUSC slides from 452 cases. After pre-processing, the mean number of patches extracted per slide at ×20 magnification is 15371.

TCGA-RCC includes three subtype projects, i.e., Kidney Chromophobe Renal Cell Carcinoma (TGCA-KICH), Kidney Renal Clear Cell Carcinoma (TCGA-KIRC) and Kidney Renal Papillary Cell Carcinoma (TCGA-KIRP), for a total of 884 diagnostic WSIs, including 111 KICH slides from 99 cases, 489 KIRC slides from 483 cases, and 284 KIRP slides from 264 cases. After pre-processing, the mean number of patches extracted per slide at ×20 magnification is 14627.

**Experiment Setup and Evaluation Metrics.**   Each WSI is cropped into a series of $256 \times 256$ non-overlapping patches, where the background region (saturation<15) is discarded. In CAMELYON16 we trained on the official training set after splitting the 270 WSIs into approximately 90% training and 10% validation, and tested on the official test set. For TCGA datasets, we first ensured that different slides from one patient do not exist in both the training and test sets, and then randomly split the data in the ratio of training:validation:test = 60:15:25. For the evaluation metrics, we used accuracy and area under the curve (AUC) scores to evaluate the classification performance, where the accuracy was calculated with a threshold of 0.5 in all experiments. For the AUC, the official test set AUC was used on the CAMELYON16 dataset, the average AUC was used on the TCGA-NSCLC dataset, and the average one-versus-rest AUC (macro-averaged) was used on the TCGA-RCC dataset. All the results over TCGA datasets are obtained by 4-fold cross-validation.

**Implementation Details.**   In the training step, cross-entropy loss was adopted, and the Lookahead optimizer [38] was employed with a learning rate of 2e-4 and weight decay of 1e-5. The size of mini-batch $B$ is 1. As in [13], the feature of each patch is embedded in a 1024-dimensional vector by a ResNet50 [39] model pre-trained on ImageNet. During training, the dimension of each feature embedding is reduced from 1024 to 512 by a fully connected layer. Finally, the feature embedding of each bag can be represented as $\mathbf{H}_i \in \mathbb{R}^{n \times 512}$. In the inference step, the softmax is used to normalize the predicted scores for each class. All experiments are done with a RTX 3090.

**Baseline.**   The baselines we chose include deep models with traditional pooling operators such as mean-pooling, max-pooling and the current state-of-the-art deep MIL models [8, 9, 13, 23, 40], the attention based pooling operator ABMIL [8] and PT-MTA [40], non-local attention based pooling operator DSMIL [9], single-attention-branch CLAM-SB[13], multi-attention-branch CLAM-MB[13], and recurrent neural network(RNN) based aggregation MIL-RNN [23].

## 4.1   Results on WSI classification

We will present the results of both binary and multiple classification. The binary classification tasks contain positive/negative classification over CAMELYON16 and LUSC/LUAD subtypes classification over TCGA-NSCLC. The multiple classification refers to TGCA-KICH/TCGA-KIRC/TCGA-KIRP subtypes classification over TCGA-RCC. All the results are provided in Table 1.

In CAMELYON16, only a small portion of each positive slide contains tumours (averagely total cancer area per slide <10%), resulting in the presence of a large number of negative regions disturbing the prediction of positive slide. The bypass attention-based methods and proposed TransMIL all outperform the traditional pooling operators. However in AUC score, TransMIL was at least 5% higher than ABMIL, PT-MTA and CLAM which neglect the correlation between instances, and

Table 1: Results on CAMELYON16, TCGA-NSCLC and TCGA-RCC.

| | CAMELYON16 | | TCGA-NSCLC | | TCGA-RCC | |
|---|---|---|---|---|---|---|
| | Accuracy | AUC | Accuracy | AUC | Accuracy | AUC |
| Mean-pooling | 0.6389 | 0.4647 | 0.7282 | 0.8401 | 0.9054 | 0.9786 |
| Max-pooling | 0.8062 | 0.8569 | 0.8593 | 0.9463 | 0.9378 | 0.9879 |
| ABMIL [8] | 0.8682 | 0.8760 | 0.7719 | 0.8656 | 0.8934 | 0.9702 |
| PT-MTA [40] | 0.8217 | 0.8454 | 0.7379 | 0.8299 | 0.9059 | 0.9700 |
| MIL-RNN [23] | 0.8450 | 0.8880 | 0.8619 | 0.9107 | \ | \ |
| DSMIL [9] | 0.7985 | 0.8179 | 0.8058 | 0.8925 | 0.9294 | 0.9841 |
| CLAM-SB [13] | 0.8760 | 0.8809 | 0.8180 | 0.8818 | 0.8816 | 0.9723 |
| CLAM-MB [13] | 0.8372 | 0.8679 | 0.8422 | 0.9377 | 0.8966 | 0.9799 |
| TransMIL | **0.8837** | **0.9309** | **0.8835** | **0.9603** | **0.9466** | **0.9882** |

do not consider the spatial information between patches. DSMIL only considers the relationship between the highest scoring instance and others, leading to limited performance.

In TCGA-NSCLC, positive slides contain relatively large areas of tumour region (averagely total cancer area per slide >80%), consequently the pooling operator can achieve better performance than in CAMELYON16. Again, TransMIL performed better than all the other competing methods, achieving 1.40% higher in AUC and 2.16% in accuracy, compared with the second best method.

In TCGA-RCC, as MIL-RNN did not consider the multi-classification problem, it was not included in this comparison result. The TCGA-RCC is unbalanced distributed in cancer subtypes and has large areas of tumour region in the positive slides (averagely total cancer area per slide >80%). However, TransMIL is equally applicable to multi-class problems with unbalanced data. It can be observed that TransMIL achieves best results in both accuracy and AUC score.

## 4.2 Ablation Study

To further determine the contribution of the PPEG module and the conditional position encoding for the performance, we have conducted a series of ablation studies. Since the high classification accuracy of most methods over TCGA-RCC is not obvious, all ablation study experiments are based on the CAMELYON16 and the TCGA-NSCLC dataset. All experiments were evaluated by AUC.

### 4.2.1 Effects of PPEG

The position encoding of the Transformer typically explores absolute position encoding (e.g., sinusoidal encoding, learnable absolute encoding) as well as conditional position encoding. However, learnable absolute encoding is commonly used in problems with fixed length sequences, and does not meet the requirement for variable length of input sequences in WSI analysis, so it is not taken into account in this paper. Here, we compared the effect of sinusoidal encoding and PPEG module which represents multi-level conditional position encoding. The same experiments are performed over CAMELYON16 and TCGA-NSCLC dataset, and the results are shown in Table 2. It should be noted that sinusoidal encoding is added to the original sequence with a multiplication of 0.001 as described in [33].

Table 2: Effects of PPEG.

| Model | Params | Camelyon16 | NSCLC |
|---|---|---|---|
| w/o | 2.625M | 0.8416 | 0.9287 |
| sin-cos | 2.625M | 0.8941 | 0.9374 |
| 3×3 | 2.630M | 0.8913 | 0.9355 |
| 7×7 | 2.651M | 0.9015 | 0.9336 |
| both | 2.669M | 0.9059 | 0.9402 |
| PPEG | 2.669M | **0.9309** | **0.9603** |

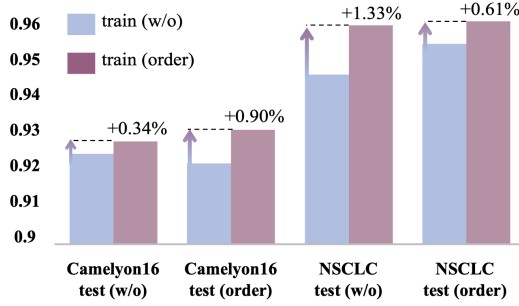

Figure 5: Effects of Positional Encoding.

Compared with the model without position encoding, it can be seen that both sinusoidal encoding and conditional position encoding can improve the classification performance, and conditional position information encoded by PPEG can be more effective in diagnosis analysis. In contrast to the $3 \times 3$ and $7 \times 7$ convolutional block, adding different sized convolution kernels in the same layer allows for multi-level positional encoding and adds more context information to each token.

### 4.2.2 Effects of Conditional Position Encoding

Here, by disrupting the order of the input sequences, we explore actual improvements for conditional position encoding. The performance of the model under different configurations is shown in Figure 5, where *order* represents sequential data input and *w/o* represents random and disordered data input. It can be seen that conditional position information did enhance the model performance, e.g., the improvement can be up to 0.9% over CAMELYON16 and 0.61% over TCGA-NSCLC in AUC.

Compared with the model without position encoding or with sinusoidal encoding, conditional position information encoded by PPEG can be more effective in diagnosis analysis. Compared with the results trained over the sequential and disordered training sets, conditional position information did enhance the model performance, e.g., the improvement can be up to 0.9% over CAMELYON16 and 0.61% over TCGA-NSCLC in AUC.

### 4.3 Interpretability and Attention Visualization

Here, we will further show the interpretability of TransMIL. As shown in Figure 6(a), the area within the blue curve annotation is the cancer region, which was provided by Gao et al. [41] over the TCGA-RCC dataset. In Figure 6(b), attention scores from TransMIL were visualised as a heatmap to determine the ROI and interpret the important morphology used for diagnosis, and Figure 6(c) is a zoomed-in view of the black square in Figure 6(b). Obviously, there is a high consistency between fine annotation area and heatmap, illustrating great interpretability and attention visualization of the proposed TransMIL.

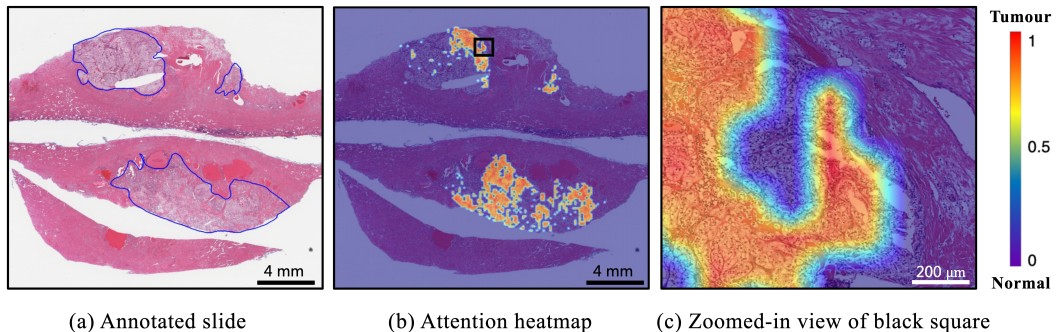

(a) Annotated slide     (b) Attention heatmap     (c) Zoomed-in view of black square

Figure 6: Interpretability and visualization.

### 4.4 Fast Convergence

Traditional MIL methods as well as the latest MIL methods such as ABMIL, DSMIL and CLAM usually require a large number of epochs to converge. Different from these methods, TransMIL makes use of the morphological and spatial information among instances, leading to approximately two to three times fewer training epochs. As shown in Figure 7, TransMIL has better performance in terms of convergence and validation AUC than other MIL methods.

## 5 Conclusion

In this paper, we have developed a novel correlated MIL framework that is consistent with the behavior of pathologists considering both the contextual information around a single area and the correlation between different areas when making a diagnostic decision. Based on this framework, a

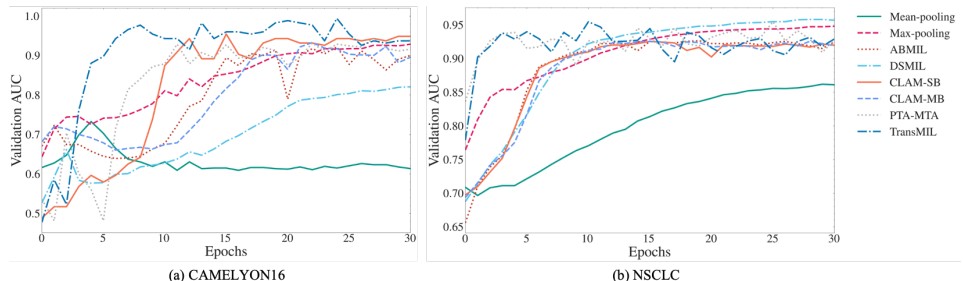

Figure 7: The convergence comparison of TransMIL and the competing methods.

Transformer based MIL (TransMIL) was devised to explore both morphological and spatial information in weakly supervised WSI classification. We also design a PPEG for position encoding as well as a TPT module with two Transformer layers and a position encoding layer. The TransMIL network is easy to train, and can be applied to unbalanced/balanced and binary/multiple classification with great visualization and interpretability. Most importantly, TransMIL outperforms the state-of-the-art MIL algorithms in terms of both AUC and accuracy over three public datasets. Currently, all the experiments were conducted over the dataset with $\times$ 20 magnification, however the WSIs with higher magnification will result in longer sequence and inevitably pose great challenges in terms of both computational and memory requirements, and we will explore this issue in the follow-up work.

**Broader Impact**   Our proposed approach shows greater potential for MIL application to real-world diagnosis analysis, particularly in problems that require more correlated information such as survival analysis and cancer cell spread detection. In the short term, the benefit of this work is to provide a model with better performance, faster convergence and clinical interpretability. In the long term, the proposed TransMIL network is more applicable to real situations, and it is hoped that it will provide more novel and effective ideas about further applications of deep MIL to diagnosis analysis.

**Acknowledgment**   This work was supported in part by the National Natural Science Foundation of China (61922048&62031023), in part by the Shenzhen Science and Technology Project (JCYJ20200109142808034), and in part by Guangdong Special Support (2019TX05X187).

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
