# A Appendix A

**Inference** *Suppose* $S : \mathcal{X} \to \mathbb{R}$ *is a continuous set function w.r.t Hausdorff distance* $d_H(\cdot, \cdot)$. $\forall \varepsilon > 0$, *for any function* $f$ *and any invertible map* $P : \mathcal{X} \to \mathbb{R}^n$, $\exists$ *function* $h$ *and* $g$, *such that for any* $\mathbf{X} \in \mathcal{X}$:

$$|S(\mathbf{X}) - g(\underset{\mathbf{X} \in \mathcal{X}}{P}\{f(\boldsymbol{x}) + h(\boldsymbol{x}) : \boldsymbol{x} \in \mathbf{X}\})| < \varepsilon. \tag{10}$$

*That is: a Hausdorff continuous function* $S(\mathbf{X})$ *can be arbitrarily approximated by a function in the form* $g(\underset{\mathbf{X} \in \mathcal{X}}{P}\{f(\boldsymbol{x}) + h(\boldsymbol{x}) : \boldsymbol{x} \in \mathbf{X}\})$.

*Proof.* As shown in Theorem 1, $\forall \varepsilon > 0$, for any invertible map $P : \mathcal{X} \to \mathbb{R}^n$, $\exists$ function $\sigma, g$, such that for any $\mathbf{X} \in \mathcal{X}$:

$$|S(\mathbf{X}) - g(\underset{\mathbf{X} \in \mathcal{X}}{P}\{\sigma(\boldsymbol{x}) : \boldsymbol{x} \in \mathbf{X}\})| < \varepsilon. \tag{11}$$

For any function $f(\boldsymbol{x})$, take $h(\boldsymbol{x}) = \sigma(\boldsymbol{x}) - f(\boldsymbol{x})$, one can derive that:

$$|S(\mathbf{X}) - g(\underset{\mathbf{X} \in \mathcal{X}}{P}\{f(\boldsymbol{x}) + h(\boldsymbol{x}) : \boldsymbol{x} \in \mathbf{X}\})| < \varepsilon. \tag{12}$$

This means $S(\mathbf{X})$ can be arbitrarily approximated by a function in the form $g(\underset{\mathbf{X} \in \mathcal{X}}{P}\{f(\boldsymbol{x}) + h(\boldsymbol{x}) : \boldsymbol{x} \in \mathbf{X}\})$.

This completes the proof. $\qquad\square$

**Theorem 2.** *The Instances in the bag are represented by random variables* $\Theta_1, \Theta_2, \ldots, \Theta_n$, *the information entropy of the bag under the correlation assumption can be expressed as* $H(\Theta_1, \Theta_2, \ldots, \Theta_n)$, *and the information entropy of the bag under the i.i.d. (independent and identical distribution) assumption can be expressed as* $\sum_{t=1}^{n} H(\Theta_t)$, *then we have:*

$$H(\Theta_1, \Theta_2, \ldots, \Theta_n) = \sum_{t=2}^{n} H(\Theta_t \mid \Theta_1, \ldots, \Theta_{t-1}) + H(\Theta_1) \leq \sum_{t=1}^{n} H(\Theta_t). \tag{13}$$

*Proof.* The Instance in the data source is represented by random variables $\Theta_1, \ldots, \Theta_n$ and the joint distribution function is $p(\theta_1, \ldots, \theta_n)$. The information entropy of data source under correlation assumption can be expressed as $H(\Theta_1, \ldots, \Theta_n)$, then we have:

$$
\begin{aligned}
H(\Theta_1, \ldots, \Theta_n) &= - \sum_{\theta_1, \ldots, \theta_n \in \vartheta^n} p(\theta_1, \ldots, \theta_n) \log p(\theta_1, \ldots, \theta_n) \\
&= - \sum_{\theta_1, \ldots, \theta_n \in \vartheta^n} p(\theta_1, \ldots, \theta_n) \log\left[p(\theta_1, \ldots, \theta_{n-1}) p(\theta_n \mid \theta_1, \ldots, \theta_{n-1})\right] \\
&= - \sum_{\theta_1, \ldots, \theta_n \in \vartheta^n} p(\theta_1, \ldots, \theta_n) \log\left\{\left[\prod_{t=2}^{n} p(\theta_t \mid \theta_1, \ldots, \theta_{t-1})\right] p(\theta_1)\right\} \\
&= - \sum_{t=2}^{n}\left[\sum_{\theta_1, \ldots, \theta_n \in \vartheta^n} p(\theta_1, \ldots, \theta_t) \log p(\theta_t \mid \theta_1, \ldots, \theta_{t-1})\right] - \sum_{\theta_1 \in \vartheta} p(\theta_1) \log p(\theta_1) \\
&= \sum_{t=2}^{n} H(\Theta_t \mid \Theta_1, \ldots, \Theta_{t-1}) + H(\Theta_1) \leq \sum_{t=1}^{n} H(\Theta_t)
\end{aligned} \tag{14}
$$

Here $\sum_{t=1}^{n} H(\Theta_t)$ is the information entropy of the data source under the i.i.d. assumption. Therefore, it is proved that the information source under the correlation assumption has smaller information entropy. In other words, correlation assumption reduces the uncertainty and brings more useful information.

This completes the proof. $\qquad\square$

# B  Appendix B

**Transformer based MIL.**  Given a set of bags $\{\mathbf{X}_1, \mathbf{X}_2, \ldots, \mathbf{X}_b\}$, and each bag $\mathbf{X}_i$ contains multiple instances $\{\boldsymbol{x}_{i,1}, \boldsymbol{x}_{i,2}, \ldots, \boldsymbol{x}_{i,n}\}$ and a corresponding label $Y_i$. The goal is to learn the mappings: $\mathbb{X} \to \mathbb{T} \to \mathcal{Y}$, where $\mathbb{X}$ is the bag space, $\mathbb{T}$ is the Transformer space and $\mathcal{Y}$ is the label space. The mapping of $\mathbb{X} \to \mathbb{T}$ can be defined as:

$$\mathbf{X}_i^0 = [\boldsymbol{x}_{i,class}; f(\boldsymbol{x}_{i,1}); f(\boldsymbol{x}_{i,2}); \ldots; f(\boldsymbol{x}_{i,n})] + \mathbf{E}_{pos}, \qquad \mathbf{X}_i^0, \ \mathbf{E}_{pos} \in \mathbb{R}^{(n+1)\times d} \quad (15)$$

$$\mathbf{Q}^\ell = \mathbf{X}_i^{\ell-1}\mathbf{W}_Q, \quad \mathbf{K}^\ell = \mathbf{X}_i^{\ell-1}\mathbf{W}_K, \quad \mathbf{V}^\ell = \mathbf{X}_i^{\ell-1}\mathbf{W}_V, \qquad \ell = 1\ldots L \quad (16)$$

$$\mathbf{head} = \mathrm{SA}(\mathbf{Q}^\ell, \mathbf{K}^\ell, \mathbf{V}^\ell) = \mathrm{softmax}\left(\frac{\mathbf{Q}^\ell\left(\mathbf{K}^\ell\right)^T}{\sqrt{d_q}}\right)\mathbf{V}^\ell, \qquad \ell = 1\ldots L \quad (17)$$

$$\mathrm{MSA}(\mathbf{Q}^\ell, \mathbf{K}^\ell, \mathbf{V}^\ell) = \mathrm{Concat}(\mathbf{head}_1, \mathbf{head}_2, \ldots, \mathbf{head}_h)\mathbf{W}^O, \qquad \ell = 1\ldots L \quad (18)$$

$$\mathbf{X}_i^\ell = \mathrm{MSA}(\mathrm{LN}(\mathbf{X}_i^{\ell-1})) + \mathbf{X}_i^{\ell-1}, \qquad \ell = 1\ldots L \quad (19)$$

where $\mathbf{W}_Q \in \mathbb{R}^{d\times d_q}$, $\mathbf{W}_K \in \mathbb{R}^{d\times d_k}$, $\mathbf{W}_V \in \mathbb{R}^{d\times d_v}$, $\mathbf{W}_O \in \mathbb{R}^{hd_v\times d}$, $\mathbf{head} \in \mathbb{R}^{(n+1)\times d_v}$, SA denotes Self-attention layer, $L$ is the number of MSA block[35], $h$ is the number of head in each MSA block and Layer Normalization (LN) is applied before every MSA block.

The mapping of $\mathbb{T} \to \mathcal{Y}$ can be defined as:

$$Y_i = \mathrm{MLP}(\mathrm{LN}((\mathbf{X}_i^L)^{(0)})), \qquad (20)$$

where $(\mathbf{X}_i^L)^{(0)}$ represents class token. The mapping of $\mathbb{T} \to \mathcal{Y}$ can be finished by using class token or global averaging pooling. Obviously, the key to Transformer based MIL is how to design the mapping of $\mathbb{X} \to \mathbb{T}$. However, there are many difficulties to directly apply Transformer in WSI classification, including the large number of instances in each bag and the large variation in the number of instances in different bags (e.g., ranging from hundreds to thousands). In this paper we focus on how to devise an efficient Transformer to better model the instance sequence.