# OpenReview forum: "TransMIL: Transformer based Correlated Multiple Instance Learning for Whole Slide Image Classification"
_NeurIPS.cc/2021/Conference — NeurIPS 2021 Poster_

### Official Review · Reviewer_XM4a · 2021-07-02

**Rating:** 6
**Confidence:** 4

**Summary:**

The paper presents a transformer based approach for WSI classification, correlation among instances are considered to address the iid assumed by previous methods. The methods were tested on three public datasets, i.e. Camelyon16, TCGA-NSCLC and TCGA-RCC datasets, and the results suggest that the performance for binary and multiple category classification is better than state of the art.

**Main Review:**


The idea of using transformr to process those instances for WSI classification seems to be an interesing idea and a good application for transformer. The authors shall include the previous work "Patch Transformer for Multi-tagging Whole Slide Histopathology Images" and discuss the main differences.  The experiments shall also include this work for comparison, as this is the closest work.

For WSI images, while image-level classfication is useful, it is also important to mark those cancer cells such that the classification can be justified. For a WSI (bag) classified as carconoma, losts of instances (patches)  included in the bag might be normal cells. How do these cells interfere with the classification results?  Can the proposed approach (transformer, correlated MIL) mitigate such issues? More detailed analysis on this problem are expected.

Minor problems:
1.	There are some spelling errors through the manuscript, e.g. the title “Classication” should be “Classification”. Please check the manuscript carefully.
2.	For the related work, the author only listed some literatures, but without giving a discussion about the strength and weakness of the existing works.
3.	In Table 1 we can see that, besides the CAMELYON16 dataset, the baseline MIL-based methods showed much lower performances than max-pooling. Please give some discussions about the reason.
4.	For ablation study, The Table 2 and Fig. 5 were not mentioned in the manuscript. What do the values stand for in Table 2 and Fig. 5? Why giving the detailed discussion in Appendix? I suggest move the discussion to the main manuscript.
5.	For Fig.6, What is the purpose to show the zoom-in view of heatmap? I cannot see anything special in this area.
6.	For Fig. 7, the initial accuracy of MIL-based baseline model were higher than the converged models, especially for the NSCLC dataset, why?


**Time Spent Reviewing:**

6

---

> ### Author Response · Authors · 2021-08-09
> **Response to Reviewer XM4a**
>
> Thank you for your invaluable feedback and comments!
>
> Kindly find below our response to the comments.
>
> - **Comparison with Li et al. 2019**: Thanks for providing this amazing work. In this paper, Li et al. have formulated the multi-head computation as an attention aggregation process to obtain enhanced feature representations for informative patch selection. We want to highlight the following differences:
>
>   - For patches, Li’s method uses the idea of multi-head module. All patch embeddings are first input into the multi-head module. Each head corresponds to one bypass attention. All the bypass attention embeddings are concatenated together, and then the dimensionality is reduced by a fully connected layer. Obviously, Li’s method did not consider the cross-patch correlation. But in our method, the TransMIL calculated attention between different patch features.
>   - In addition to correlation information, Li’s paper did not consider the spatial information between patches. But we design the PPEG module to explore the spatial information.
>   - We hope this highlights our unique contribution from Li’s method, and we will add this work as a competing method in the experiment of our paper. We reproduced the code of Li’s method in our experiment. The performance is illustrated in the following table, and we can still observe significant performance gains. We will add this experiment result in the manuscript.
>
>   |      | Method                   | Camelyon16-Acc | Camelyon16-AUC | NSCLC-Acc  | NSCLC-AUC  | RCC-Acc    | RCC-AUC    |
>   | ---- | ------------------------ | -------------- | -------------- | ---------- | ---------- | ---------- | ---------- |
>   | 1    | PT-3head-MTA (Li et al.) | 0.8217         | 0.8454         | 0.7379     | 0.8299     | 0.9059     | 0.9700     |
>   | 2    | TransMIL (Ours)          | **0.8837**     | **0.9309**     | **0.8835** | **0.9603** | **0.9466** | **0.9882** |
>
> - **False positive problem**: For weakly supervised WSI classification, false positive problem is one of the most important elements in terms of classification accuracy. Previous works have adopted the attention method to increase the attention weight of cancer areas while reducing the weight of non-cancer areas to alleviate this problem. But they ignore the correlation and spatial information between patches. We believe this is very important for accurate attention information. So, we designed a self-attention based TransMIL in this paper, and many experimental results verified our ideas. Especially, over the Camelyon16 dataset (averagely total cancer area per slide <10%), it can be seen from Table 1 in this paper that our TransMIL can also achieve the best results for the cancer with small tumor area. It shows that TransMIL can focus more accurately on the cancer area than previous Attention method, thereby reducing the false positive rate.
>
> - **Ablation Study**:
>   - Due to space limitations, we put the detailed analysis of the ablation experiment in the appendix, and we will put it in the paper.
>   - Table 2 includes the ablation studies that explore the model's performance in absolute position encoding and conditional position encoding. *sin-cos* means the sinusoidal encoding is added to the original sequence with a multiplication of 0.001, 3$\times$3, 7$\times$7 and *PPEG* means convolutional position coding of different parameters, *both* means sinusoidal encoding and conditional position encoding are both added. *PPEG* means Transformer-PPEG-Transformer and it achieves the best result.
>   - In Figure 5, by disrupting the order of the input sequences, we explore actual gains for conditional position encoding in PPEG. where *order* represents sequential data input and *w/o* represents random and disordered data input. Conditional position information did enhance the model performance.
> - **Table 1**: In TCGA-NSCLC, positive slides contain relatively large areas of tumor region (averagely total cancer area per slide >80%). In this paper, the baseline MIL-based methods indeed showed much lower performances than max-pooling. We carefully rechecked all the original data and found that the result of Max-pooling method has no problem. We believe the possible reasons are as follows: 1. In order to ensure the fairness of the experiment, we set the same training hyperparameters for each competing model. One potential possibility is that the Max-pooling method is more robust for the hyperparameters setting, so it is better than the baseline method. 2. The Max-pooling method, to a certain extent, can avoid the interference caused by the negative patch in the positive slide from some perspectives. However, the competing attention-based methods may not effectively pay attention to the true positive area, and lead to lower performance than the Max-pooling method.
> - **Fig 6**: The purpose of showing the zoom-in view of heatmap is as follows: We want to show that the heat map and the tumor area can roughly correspond well in local details to prove that TransMIL can get the accurate attention.
>
> - **Literature Review**: Thank you for pointing us this problem. We will give a discussion about the strength and weakness of the existing works.
>   - For the application of MIL in WSI classification, instance-level algorithms select top-k key instances in the slide to represent the slide label. For some cancers with small tumor areas, this method can reduce false positive rate. However, this method requires many WSIs, since only several instances within each slide can actually participate in the training. The embedding-level method can make use of all the information in the entire slide, and the commonly used method is attention-based MIL, but proposed methods did not consider the correlated and spatial information between different instances.
> - **Fig.7** : Thank you for pointing us this problem. We carefully rechecked all the original data and found that the initial value of the DSMIL method in the NSCLC dataset was incorrectly saved. The reasons are as follows: The pytorch-lightning(pl) framework we used has a mechanism that pl will first check the model by some batches and then calculate and save the Val ACC(in fact, it should be removed) before training the real model. When we processed the data of DSMIL, we forgot to delete the useless Val ACC(0.8741, higher than the final converged value). The correct ACC value after the first epoch should be 0.6071, which is lower than final converged value. We are terribly sorry for that and we will update Fig. 7 (b) in the manuscript. We promise to open source the code on github to ensure the reproducibility of the results.
> - **Spelling errors**: We apologize for these spelling errors and will check and revise the manuscript carefully in the future.

---

### Official Review · Reviewer_Bp25 · 2021-07-11

**Rating:** 7
**Confidence:** 4

**Summary:**

	The authors propose a Transformer-based MIL method for WSI classification.
The method is designed to avoid the IID assumption inherent to standard MIL inference methods (e.g. mean, max pooling). Secondly, they propose a positional encoding module for patches in WSIs based on CNNs to incorporate spatial information into the prediction, along with semantic information learned in the MSA layers. In the ablation study, they demonstrated that PPEG model performed better than conventional sin-cos positional encoding. They used three WSI classification datasets from open sources with previously benchmarked results. The authors were able to achieve SOTA results on each task.


**Limitations And Societal Impact:**

Weakness:
One major problem, or point of misunderstanding, is how the ‘Spatial Restore’ step in Algorithm 3 PPEG processing flow is related to the actual spatial relationships between patch locations in the WSI. The input order of patches into the MSA module is unclear from the paper, (random, sliding window, etc. ?), but there is no ‘natural’ square in the WSI. Yet in processing the patch tokens (H_f), an artificial 2-d square feature map is generated. With the patch tokens now populating the ‘pixels’ in the feature maps, adjacent patch tokens/pixels are not necessarily locally connected and ought not to belong in the same convolutional receptive field. I assume this is what the authors mean by ‘context information', however, it is unclear how the PPEG module provides the appropriate patch-level spatial context within the WSI. Different sized convolutions do not address this problem.

If the above criticism is correct, does the model perform differently when patch tokens are randomly shuffled versus structured sliding-window patching (semi-ordered)? If they do not, then PPEG is learning to perform some other function than incorporating spatial/context information.

Adding attention maps from CLAM versus ABMIL versus TransMIL to Figure 6 would be constructive. As proposed, it is unclear what is actually driving the boost in performance: better spatial/bypass attention or correlative/self-attention.


**Main Review:**

Originality: original work

Quality: very well done, figures, tables, algorithms look excellent

Clarity: The paper is dense and the authors attempted to cover a lot of ground. The writing is clear for the space alotted.

Significance: Paper is significant for moving WSI classification forward by applying Tranformers to MIL.

Strengths:
This paper is the natural extension of previous RNN-based and attention-based MIL methods for WSI classification in computational pathology. The application of Transformers to patch-based WSI classification is a natural one given the recent success of Transformers for NLP (patches-as-token).
The authors selected three representative WSI datasets for model testing.
Figure 2 is excellent and provides a clear illustration of the difference between the MIL inference methods using an explicit pooling matrix.
The authors are commended for stating explicitly, and addressing, that an IID assumption for patch-based WSI classification is incorrect and should be avoided in the development of models moving forward.


**Time Spent Reviewing:**

3

---

> ### Author Response · Authors · 2021-08-09
> **Response to Reviewer Bp25**
>
> Thank you for your invaluable feedback and comments!
>
> Kindly find below our response to the comments.
>
> - **Input order**: As gigapixel images, WSIs have a large amount of blank background, so we first discard the background patches. And then for the remaining patches, the input sequence is composed from left to right in the raster scanning order.
>
> - **Spatial information:** To better retain the 2D positional information of the patches in WSI, we used convolution to encode conditional position information. Considering the variable length of input sequence in WSI analysis, we make an artificial 2D square feature map to facilitate the processing of convolution. Since the input sequence is composed from left to right in the raster scanning order, the proposed method keeps more of the left and right adjacent information of each patch, that is the 1D spatial context under the X axis. Under this scanning manner, part of the Y-axis position information may be lost, but it is still better than the disordered sequence. Overall, what we proposed is a feasible but may be not the most optimal solution to this tricky problem and we will continue to explore this problem in the future.
>
> - **Context** **information:** The context information we represent in the manuscript is relative to the squared feature map, so different sized convolutions can obtain different aggregated information. Of course, what the reviewer described is another interpretation of context information in terms of the real space level.
>
> - **Boost in performance**: We believe both PPEG and correlative/self-attention are driving the boost in performance. 1. For PPEG, it plays the role of spatial/context information aggregation. The experiment result shows that it performs differently when patch tokens are randomly shuffled versus structured sliding-window patching (semi-ordered). For example, in Fig. 5,  the model trained in ordered sequence can obtain better performance than disordered sequence. 2. For SA, it plays the role of correlative/self-attention. And for the effectiveness of the correlative/self-attention, we added ablation experiments in the response to GWgS.
>
> - **Attention maps**: Thanks for pointing us this problem, attention maps from CLAM versus ABMIL versus TransMIL will be added.

---

> > ### Comment · Reviewer_Bp25 · 2021-09-02
> > **Response to Authors**
> >
> > The authors have addressed my concerns. I have upgraded my recommendation.

---

### Official Review · Reviewer_GWgS · 2021-07-28

**Rating:** 6
**Confidence:** 3

**Summary:**

The authors propose a transformer based model for the classification of whole slide histopathology images. The method first extract ResNet features locally across WSIs, then apply a transformer based model on top of the features. The authors also propose a Pyramid Position Encoding Generator (PPEG) module for transforming local features and encode positional information. The proposed framework is general and can be applied to other non-medical image analysis tasks.

The proposed method achieves better result compared to recent baselines.

**Limitations And Societal Impact:**

The authors can show failure cases and discuss possible improvements.

**Main Review:**

The theorems give some analysis on why multi-head self-attention layers may be good for this application. However, conclusion of "smaller information entropy" is not exciting. To some extent, it is almost straightforward to think that adding cross-patch correlation is beneficial. The paper is somewhat hard to follow. The authors introduced terms like TPT and PPEG before these terms were explained.

I have a couple of concerns on experiments:
1. The authors claim that the multi-head self-attention layer models the correlation across patches. To me, the convolutional layers in the PPEG module also mix features across patches, and thus, uses spatial patterns (correlation) across patches. It is not clear to me if simply stacking multiple PPEG modules would perform similarly, compared to the proposed approach. It would be great if authors can show the effectiveness of the attention layers via ablation studies.
2. The authors compared their proposed PPEG against commonly used positional encoding methods such as sin-cos. It is not clear to me if the comparison is fair. First, without any non-linearity layer between self-attention layers, the self-attention layers tend to converge to a trivial state of all tokens having the same feature vector. Thus, if the authors use sin-cos as the baseline method, the authors should explain if they still put some layers with non-linearity in the place of PPEG. Second, PPEG introduces computation and increases model capacity, where as sin-cos barely does. It is not clear if the performance gain is due to increased model capacity, or due to the intended mechanism.

Minor problems:
1. What does "TPT" stand for?
2. "discraded" should be "discarded" in the caption of Figure 3.

**Time Spent Reviewing:**

1.5

---

> ### Author Response · Authors · 2021-08-09
> **Response to Reviewer GWgS**
>
> Thank you for your invaluable feedback and comments!
>
> Kindly find below our response to the comments.
>
> - **Theorems**: Theorem 1 and 2 are integral and continuous theory, they constitute the theoretical basis of correlated MIL together. Theorem 1 and its inference are the core of these theoretical proofs, since they prove that it is mathematically feasible to break the i.i.d assumption of MIL, and it is also the prerequisite for Theorem 2. Theorem 2 further illustrates the validity of the correlated MIL from the perspective of information entropy. We will add the explanation of terms like TPT and PPEG before introducing such terms to increase the readability of this paper.
>
> - **Ablation Study of Self-attention Layer**: To better illustrate the role of self-attention (SA) layer in model performance improvement, we set up the following ablation experiment over Camelyon16 dataset.  It can be seen from the results of lines 1, 2, and 3 that simply stacking PPEG cannot achieve good results.  PPEG can encoder local feature and SA has a global receptive field, that the combination of this structure can simultaneously aggregate local and global information, thereby improving the performance of the model. From lines 1,2,3,7, we can see structure of SA+PPEG has  good performance, and we believe this is because it can greatly broaden the receptive field of the model . As shown in line 8, the latter SA layer further aggregates feature tokens, and extracts classification information through class token. This is why the proposed SA+PPEG+SA can achieve the best result in all the experiments.
>
>   |      | Model                              | AUC                  |
>   | ---- | ---------------------------------- | -------------------- |
>   | 1    | 1$\times$PPEG + Attn/Mean/Max      | 0.7890/0.6645/0.8298 |
>   | 2    | 5$\times$PPEG + Attn/Mean/Max      | 0.6888/0.6416/0.7395 |
>   | 3    | 10$\times$PPEG + Attn/Mean/Max     | 0.5750/0.6665/0.6651 |
>   | 4    | 1$\times$PPEG + SA                 | 0.5413               |
>   | 5    | SA                                 | 0.8579               |
>   | 6    | SA + SA                            | 0.8416               |
>   | 7    | SA + 1$\times$PPEG + Attn/Mean/Max | 0.9186/0.8875/0.9122 |
>   | 8    | SA + 1$\times$PPEG + SA            | 0.9309               |
>
>
>
> - **Non-linearity layer and model capacity**:
>
> In response to reviewer's concerns about non-linearity layer and model capacity, we conducted the following experiments.
>
> To highlight the effectiveness of the PPEG, we add different non-linearity layers in sin-cos experiment, including Feed Forward Network (FFN) and PPEG. The implementation of FFN includes ReLU, fully connected layer and Layer Norm. The results in line 2~5 show that PPEG can be replaced by some non-linearity layers in sin-cos experiment, but there is no performance gain like PPEG.
>
> 5$\times$/10$\times$PPEG represent 5/10 PPEGs stacked together respectively, that is T(5$\times$PPEG)T and
>
> T(10$\times$PPEG)T. The results in line 6~8 show the performance gain is not due to the increased model capacity but the intended mechanism.
>
> |      | Model                    | Params | CAMELYON16 |
> | ---- | ------------------------ | ------ | ---------- |
> | 1    | w/o                      | 2.6M   | 0.8416     |
> | 2    | sin-cos                  | 2.6M   | 0.8941     |
> | 3    | sin-cos+FFN              | 4.7M   | 0.8640     |
> | 4    | sin-cos+FFN$\times$2     | 6.8M   | 0.8972     |
> | 5    | sin-cos+PPEG             | 2.7M   | 0.9059     |
> | 6    | T(5$\times$PPEG)T        | 2.9M   | 0.9151     |
> | 7    | T(10$\times$PPEG)T       | 3.1M   | 0.9189     |
> | 8    | T(1$\times$PPEG)T (ours) | 2.7M   | **0.9309** |
>
> - **TPT**: TPT stands for our proposed Transformer layer - PPEG - Transformer layer structure. We will give detail explanation in the manuscript.
> - **Spelling errors**: We apologize for these spelling errors and have checked the manuscript again.

---

> > ### Comment · Reviewer_GWgS · 2021-08-31
> > **Concerns Cleared**
> >
> > I have read all reviewer feedbacks and the authors' responses. My concerns are cleared. I upgraded my ratings to "marginally above the acceptance threshold".
> >
> > I encourage the authors to discuss the work of [Li et al. 2019], and compare the proposed method to it, if this paper is accepted (as suggested by the other reviewer). In addition, if space allows, the authors can include experiment results of adding non-linearity layers with sin-cos.

---

### Decision · Program_Chairs · 2021-09-27

**Decision:**

Accept (Poster)

**Comment:**

The paper has an interesting application (if rather straightforward) of transformers on Multiple Instance Learning for classification of Whole slide images. The authors successfully responded to the reviewer criticisms. All reviewers recommend acceptance.

Authors are encouraged to include the additional experiments and explanations to the final version